# In-Depth Analysis of the Pancreatic Extracellular Matrix during Development for Next-Generation Tissue Engineering

**DOI:** 10.3390/ijms241210268

**Published:** 2023-06-17

**Authors:** Laura Glorieux, Laura Vandooren, Sylvie Derclaye, Sébastien Pyr dit Ruys, Paloma Oncina-Gil, Anna Salowka, Gaëtan Herinckx, Elias Aajja, Pascale Lemoine, Catherine Spourquet, Hélène Lefort, Patrick Henriet, Donatienne Tyteca, Francesca M. Spagnoli, David Alsteens, Didier Vertommen, Christophe E. Pierreux

**Affiliations:** 1Cell Biology Unit, de Duve Institute, UCLouvain, 1200 Brussels, Belgium; 2Nanobiophysics Lab, Louvain Institute of Biomolecular Science and Technology, UCLouvain, 1348 Louvain-la-Neuve, Belgium; 3Louvain Drug Research Institute, UCLouvain, 1200 Brussels, Belgium; 4Centre for Gene Therapy and Regenerative Medicine, King’s College London, Great Maze Pond, London SE1 9RT, UK; 5de Duve Institute and MASSPROT Platform, UCLouvain, 1200 Brussels, Belgium

**Keywords:** pancreas, extracellular matrix, development, decellularization, proteomics, atomic force microscopy

## Abstract

The pancreas is a complex organ consisting of differentiated cells and extracellular matrix (ECM) organized adequately to enable its endocrine and exocrine functions. Although much is known about the intrinsic factors that control pancreas development, very few studies have focused on the microenvironment surrounding pancreatic cells. This environment is composed of various cells and ECM components, which play a critical role in maintaining tissue organization and homeostasis. In this study, we applied mass spectrometry to identify and quantify the ECM composition of the developing pancreas at the embryonic (E) day 14.5 and postnatal (P) day 1 stages. Our proteomic analysis identified 160 ECM proteins that displayed a dynamic expression profile with a shift in collagens and proteoglycans. Furthermore, we used atomic force microscopy to measure the biomechanical properties and found that the pancreatic ECM was soft (≤400 Pa) with no significant change during pancreas maturation. Lastly, we optimized a decellularization protocol for P1 pancreatic tissues, incorporating a preliminary crosslinking step, which effectively preserved the 3D organization of the ECM. The resulting ECM scaffold proved suitable for recellularization studies. Our findings provide insights into the composition and biomechanics of the pancreatic embryonic and perinatal ECM, offering a foundation for future studies investigating the dynamic interactions between the ECM and pancreatic cells.

## 1. Introduction

Tissue engineering has the potential to revolutionize regenerative medicine, but it requires a deep understanding of the 3D organization of the organ one wishes to replicate and the bioavailability of viable and mature cells. Indeed, the composition, organization, and mechanical properties (i.e., viscoelasticity) of each organ are unique and cannot be generalized. Moreover, even with current protocol improvements, obtaining fully differentiated cells remains difficult. One potential solution is to use progenitor cells or not fully differentiated cells in a flexible or plastic embryonic matrix and allow them to differentiate and remodel the matrix [1]. To achieve this goal, it is crucial to know the composition and elucidate the mechanical properties of the embryonic matrix, in order to ultimately use its structure to orient the function of progenitor cells.

The pancreas, with its complex organization and dual endocrine and exocrine functions, is a particularly challenging target. During development, the pancreas undergoes a multi-step process involving the proliferation, differentiation, and morphogenesis of various cell types, including acinar, ductal, and endocrine progenitors, all deriving from the endoderm [2,3]. This process is governed by intra- and extra-cellular cues from the surrounding connective tissue composed of endothelial cells, nerve cells, mesenchymal cells, fibroblasts, and extracellular matrix (ECM) components. Crosstalk between pancreatic epithelial cells and mesenchymal or endothelial cells has been investigated during pancreatic development [4]. Ablation of mesenchyme-derived cells causes pancreas agenesis, while the absence or decreased abundance of endothelial cells prevents early pancreas budding and, later, promotes acinar differentiation [5,6]. However, the influence of ECM proteins and structural properties on pancreas organogenesis is still poorly understood and deserves further investigation.

The ECM is a complex dynamic network of macromolecules, synthesized and degraded by cells, which is essential to support, organize, and hydrate tissues and to confer elasticity and compliance upon deformation [7]. Moreover, ECM components can interact with growth factors and cell surface receptors to regulate cellular biochemical and mechanical processes, including cell proliferation and differentiation [8]. For instance, laminin111, a well-studied ECM protein, is detected as early as Embryonic day (E) 10.5 in the pancreas, whose abundance is not homogeneous around the pancreatic epithelial cells. It can stimulate tubulogenesis and repress acinar differentiation [9,10]. Besides composition, ECM elasticity has also been shown to impact ductal or endocrine fate. If allowed to spread and interact with the ECM, pancreatic progenitors differentiate into ductal cells via F-actin/YAP-1/Notch signaling, while if confined, progenitors undergo endocrine differentiation [11]. Given the number of proteins forming the matrix and the few studies on their role in development, it is of utmost importance to better characterize and understand this extracellular compartment.

Recently, research on the ECM has gained significant attention in the field of tissue engineering. Due to the ECM’s multifaceted biochemical complexity, structural specificity, and tissue-specific functionalities, it could exert a profound influence on cell fate and differentiation [12]. An ECM scaffold can be obtained by eliminating cells from an organ through chemical, physical, or enzymatic methods, while preserving the essential characteristics of the ECM, encompassing biochemical and biomechanical parameters [13]. Despite their successful application in various organs, such as the adult pancreas, the precise impact of ECM hydrogels on cell differentiation and maturation, in particular endocrine cell differentiation for diabetes, remains unknown and understudied [14,15]. The same holds true for the mechanical properties of the ECM. In addition, the composition and mechanical properties of the adult pancreatic ECM may not be as informative as its embryonic pancreatic counterpart, if one wants to improve pancreatic cell differentiation protocols, which occurs primarily during embryogenesis.

In this study, we conducted an in-depth analysis of the composition and mechanical properties of the pancreatic ECM during development, with a focus on two key stages: E14.5, when the pancreas is an immature tissue populated with acinar, ductal, and endocrine progenitor cells, and Postnatal day 1 (P1), when the pancreas is a fully differentiated and functional organ. Despite the small size of the embryonic tissue, we successfully adapted available protocols to collect data and compare the ECM components at both the embryonic and postnatal stages. By implementing an additional fractionation step, we were able to significantly increase the number of proteins detected. To investigate the mechanical characteristics, i.e., elasticity and viscosity of the pancreas at these stages, we employed atomic force microscopy. By subtracting the contributions of pancreatic cells, we were able to determine the mechanical properties solely attributable to the ECM. Innovatively, we devised a novel decellularization protocol specifically designed for embryonic and postnatal pancreas, which effectively removed cells, while preserving the three-dimensional (3D) structure. Notably, we introduced a crosslinking step prior to decellularization, resulting in improved maintenance of the structural organization of the 3D ECM, as compared to non-crosslinked decellularized ECM. To demonstrate the biocompatibility of our pancreatic ECM scaffold, we seeded embryonic stem cells and reported viability. The novel decellularization protocol and the newly acquired knowledge regarding the biochemical and biomechanical properties of the embryonic ECM should promote and guide the development of engineered ECM for tissue engineering. 

## 2. Results

### 2.1. In-Depth Proteome-Wide Identification and Quantification of Pancreatic Matrisome

Liquid chromatography coupled with tandem mass spectrometry (LC-MS/MS) has emerged as the gold standard technique to determine the proteomic composition of a tissue. The label-free quantification, which consists of measuring precursor ions’ intensities, is routinely used for quantification and comparison of proteins in biological tissue. Here, we adapted a proteomic workflow [16], developed to allow identification and quantification of ECM proteins, to small E14.5 and P1 pancreatic tissues. Briefly, we collected and lysed 5 mg (wet weight) of pancreatic tissue with the help of steel beads in a tissue homogenizer (Figure 1A, Step 1). This first mechanical lysis resulted in the separation of the sample into two fractions: the supernatant, enriched in cellular and matrisome-associated proteins, and the pellet, enriched in core matrisome proteins. The pellet fraction (Fraction 2 in Figure 1A) was further disrupted with enzymes and solubilized in urea to release the entangled proteins (Figure 1A, Step 2). After methanol/chloroform precipitation, proteins of Fraction 1 were digested into peptides with trypsin, while proteins from Fraction 2 were first digested, followed by the purification and concentration of the peptides using a desalting column (Figure 1A, Step 3). Finally, peptides were fractionated into six additional fractions using reverse-phase chromatography at a high pH to reduce sample complexity, thereby improving proteome and low-abundant protein detection by LC-MS/MS (Figure 1A, Step 4). Proteins were identified and quantified using Proteome Discoverer 2.5 against a mouse protein database. Label-free quantification (LFQ) using peak areas was performed on the peptides’ precursor intensities (AUC).

In total, we confidently identified 5169 proteins (with a false discovery rate of less than 1%, a minimum of three peptide spectral matches, and at least two unique peptides per protein detected). Out of these, 167 were categorized as matrisome proteins, defined as the ensemble of genes encoding ECM and ECM-associated proteins, based on an in silico database of the Matrisome project (Figure 1B) [17]. The matrisome was further categorized into core matrisome proteins (including ECM glycoproteins, collagens, and proteoglycans) and matrisome-associated proteins (including ECM-affiliated proteins, ECM regulators, and secreted factors). To refine our results, we filtered out proteins that were not detected at least three times in the experiments (out of six samples, three at E14.5 and three at P1), leading to a total of 4869 proteins including 160 matrisome proteins. Out of the 160 detected and quantifiable proteins, all were detected at both stages and classified into 15 collagen types, 54 glycoproteins, 6 proteoglycans, 46 ECM regulators, 30 ECM-affiliated proteins, and 9 secreted factors (Figure 1B).

Next, we performed a principal component analysis on all quantifiable proteins to investigate batch effects and sample differences. Overall, biological replicates appeared to separate by group, into either the E14.5 or P1 developmental stage, but a weak batch variability was observed (Appendix A). Nevertheless, visualization of the biological samples with a heatmap of normalized matrisome proteins suggested larger intergroup differences than intragroup variations (Appendix A). 

### 2.2. Differential Matrisome Signature between Embryonic (E) Day 14.5 and Postnatal (P) 1 Pancreas by Label Free Quantification (LFQ)

To investigate the composition and dynamics of the ECM during pancreatic development, we conducted a paired *t*-test on all quantifiable proteins and compared the distribution of core matrisome proteins between the E14.5 (*y*-axis) and P1 (*x*-axis) developmental stages (Figure 2A). Our analysis revealed the presence of various collagen types, including fibrillar (COL1A1, COL1A2, COL3A1, COL5A1, and COL5A2), basement membrane (COL4A1, COL4A2, COL4A3, and COL4A6), filamentous (COL6A1, COL6A2, and COL6A3), fibril-associated with interrupted triple helices (COL12A1 and COL14A1), and multiplexins (COL15A1 and COL18A1) in both stages. Except for the COL1A2 isoform, all other collagen types were found to be more abundant at P1 as compared to E14.5. In contrast, most of the detected proteoglycans (DCN, BGN, VCAN, and OGN) were more abundant in E14.5 samples. We performed the same comparative analysis for ECM glycoproteins and observed that, similar to the collagens, the majority of glycoproteins were found to be more abundant in the P1 samples. However, some proteins, such as agrin (AGN), EMILIN1, TGFBI, and spondin1 (SPON1), were constant throughout development (Figure 2A). To gain a global picture of the dynamic profile of the ECM at both stages, similar plots were created for the categories of matrisome-associated proteins, including ECM-regulators, ECM-affiliated proteins, and secreted factors (Appendix A).

To identify significant protein changes between E14.5 and P1 pancreatic tissue, we represented the results in a volcano plot (adj. *p* value < 0.05, difference 1.5). The plot revealed that 27 core matrisome proteins were significantly enriched in P1 pancreatic tissue, including all collagen IV and VI isoforms, LAMA2, LAMA5, VTN, and DMBT1 (Figure 2B (left) and Appendix A). Only two core matrisome proteins, BGN and DDX26B, were significantly enriched in E14.5 pancreatic tissue (Figure 2B (left)). Regarding the matrisome-associated proteins, 25 were significantly enriched in the P1 pancreas and only 3 at E14.5 (Figure 2B (right)). Higher expression of proteins related to exopeptidase produced by the exocrine pancreas (TRY5, TRY10, PRSS2, and PRSS3) and considered as ECM regulators were observed in the P1 samples, which coincides with the more mature state of the pancreas.

Finally, we investigated the relative abundance of proteins at E14.5 and P1 using peptide spectral matches (PSMs) and ranked the top ten core matrisome proteins based on their PSM counts (Figure 2C). PSMs provide valuable information about the presence or absence of peptides corresponding to specific proteins in the sample. This revealed a majority of proteins related to the basement membrane at E14.5 (LAMB1, HSPG2, LAMA5, LAMC1, and FN1), while interstitial ECM became more abundant at P1 (COL6A1, COL6A2, COL6A3, and COL12A1). Overall, our results demonstrated a distinct matrisome signature at E14.5 and P1, characterized by an increase in collagenous proteins and glycoproteins, accompanied by a decrease of proteoglycans through development. 

### 2.3. Constant Elasticity and Viscosity Properties of Pancreatic Extracellular Matrix (ECM) during Development

It is well established that mechanical cues from the ECM can influence cell statuses, such as cell proliferation and differentiation [18,19]. These cues result from the composition, degree of crosslinking, and density of the ECM components, and can be measured through the apparent elasticity (Young’s modulus E) and viscosity of the ECM [20]. 

To assess the elasticity of the pancreatic ECM, we employed the atomic force microscopy (AFM) micro-indentation technique. We probed fresh micro-dissected E14.5 and P1 pancreatic tissues and indented them with a spherical bead of 6 µm in diameter to minimize tissue damage and averaged the value on a larger area (Figure 3A) [21]. All measurements were performed with an indentation force of 5 nN, which corresponds to a depth of a few micrometers, ensuring that the substrate effects were negligible. By analyzing the cantilever’s deflection during tissue indentation, we obtained a force–displacement curve and analyzed the approach curve to determine Young’s modulus (for further details, please refer to the Section 4). Our results showed that the elasticity of pancreas ranged from 100 to 2000 Pa, with no significant difference between the two stages (Figure 3B). When probing fresh pancreatic tissues, we assumed that we mostly encountered the ECM of the micro-dissected tissue. However, we probably also encountered surrounding pancreatic mesenchymal cells or fibroblasts, as suggested by the non-uniform distribution of elasticity in the maps at both stages (Figure 3D). 

To evaluate this possibility, we micro-dissected and dissociated the P1 pancreas to assess the elasticity of isolated cells using the same technology (Figure 3A’). The mesenchymal/fibroblastic nature of these cells was confirmed by the presence of VIMENTIN and the absence of E-CADHERIN staining (Appendix A). The results showed that the elasticity of the VIMENTIN+ cells was higher with values between 425 and 4000 Pa (Figure 3B). We ruled out the effect of glass dishes underneath that showed higher elasticity (around 600 kPa) by limiting the analysis to an indentation depth of 500 nm. This result suggested that values below 425 found in the pancreatic tissues could be attributed to the sole ECM. By eliminating the possible impact of elasticity from the cellular compartment of the tissue, we propose that the overall elasticity of the pancreatic matrix at E14.5 and P1 was relatively low, ranging from 100 to 400 Pa at both stages (Figure 3C). The absence of difference between the two stages implied that elasticity remained consistent throughout development. This conclusion was backed up by the examination of earlier developmental stages, such as the E12.5 pancreas, which again showed no noticeable differences and similar elasticity maps (Appendix A).

The presence of hysteresis between the indentation and retraction curve, observed in the force–displacement measurements, is a typical feature of viscoelastic behavior. This observation prompted us to investigate the viscosity of pancreatic tissue throughout development. To this end, we employed a numerical model developed by Abuhattum et al. [22], which consisted of fitting classical AFM force–indentation curves with equations derived from two viscoelastic mechanical models, Kelvin–Voight and Maxwell. This process allows the computation of viscosity parameters on already acquired data without the need for additional time-consuming and invasive experiments. We applied this model on the AFM curves from the E12.5, E14.5, and P1 pancreas and obtained four parameters, namely the apparent elasticity that corresponds to the previously calculated elasticity, the unrelaxed elasticity, the Maxwell relaxation time, and the viscosity (Appendix A). By excluding the results with apparent elasticity above 425 Pa, we computed a low viscosity of the pancreatic ECM with values around 0 to 2 Pa.s for both stages (Figure 3E). While not statistically significant, we observed a slight increase in pancreatic ECM viscosity over time.

### 2.4. Preservation of Decellularized ECM Scaffold Organization by Crosslinking

For the ease of tissue collection and manipulation, as well as the larger size of the tissue, we continued to work on P1 samples. The goal was to decellularize pancreatic tissue in order to (1) obtain a cell-free ECM scaffold, (2) retain a significant amount of matrisome proteins, and (3) maintain its 3D structure. We used a NaOH-based protocol [23], which effectively removed cells, as evidenced by the absence of hemalun, eosin, fast green, DAPI, and E-CADHERIN stainings and the preserved key structural proteins such as collagen, hyaluronic acid, and laminin (compare the first row (P1 pancreas) and second row (dECM) of Figure 4 and Appendix A). However, Alcian blue staining revealed a relative loss of glycosaminoglycans, as compared to the P1 pancreas. More importantly, we observed the degradation of the dECM macro-architecture, which appeared to collapse and presented a discontinuous basement membrane (laminin and collagen IV), as compared to the native P1 pancreatic tissue (Figure 4). 

In an attempt to maintain glycosaminoglycans and preserve the 3D structure of the matrix, we incubated P1 pancreas with a crosslinking agent 1-ethyl-3-(3-dimethylaminopropyl) carbodiimide (EDC) prior to the decellularization process. This particular crosslinker was chosen for its ability to catalyze a covalent reaction between carboxylic acid and amine groups. Furthermore, the molecular properties of EDC prevent it from penetrating the cell membranes, thereby ensuring that the crosslinking reactions occur exclusively outside of the cells within the ECM. We experimentally determined the ideal conditions: 5 mM of EDC for a duration of 5 min (for the detailed procedures, please refer to the Section 4). For the sake of clarity, we refer to the crosslinked and decellularized pancreatic ECM scaffold as c+dECM (last row of Figure 4). The crosslinking approach effectively preserved the ECM structure, as shown by the resemblance with the native tissue without preventing cell removal by NaOH (Figure 4; please compare the P1 pancreas with c+dECM; Appendix A). Immunofluorescence labelling confirmed the preservation of the ECM macrostructure in c+dECM, in particular the organization of the basement membranes (LAMININ and COLLAGEN IV) of epithelia and endothelia. Further validation of the preservation of the 3D structure was performed using light sheet microscopy and two-photon microscopy (Figure 5 and Appendix A). Light sheet microscopy revealed a similar organization of the basement membrane, visualized with COLLAGEN IV and LAMININ labelling, in native pancreatic tissue and c+dECM (Figure 5). However, despite the use of this crosslinking agent, most sulfated glycosaminoglycan components (sGAG) were lost in the c+dECM scaffold (Appendix A). It should be noted that this EDC crosslinking step was crucial for the maintenance of the ECM’s 3D structure when decellularizing tissues at earlier stages of pancreas development such as E14.5 (Appendix A), but less important at more advanced postnatal stages such as P14 (Appendix A). 

### 2.5. Crosslinked and Decellularized ECM Scaffold Is Biocompatible and Allows Cell Survival

To investigate whether the postnatal decellularized pancreatic matrix is a cell-compatible biomaterial, and in particular can sustain embryonic cell viability, we incubated cells with the P1 pancreatic decellularized ECM scaffolds. We placed the dECM and c+dECM scaffolds on a semi-porous filter floating on the culture medium. Then, one-million mouse embryonic stem cells (E14Tg2A) were seeded on top of the ECM scaffold on the filter, and the cell-ECM scaffold mixture was cultured at the air–medium interface for 6 days. 

Cell penetration into the scaffold and cell viability were evaluated by histological staining and immunolabelling of the seeded ECM scaffolds at the end of the culture (Figure 6A). The low magnification image of H&E staining first revealed that cells remained predominantly around the ECM scaffold, being unable to penetrate the interstitial matrix or the empty acinar and ductal spaces of the decellularized pancreatic ECM. However, cell penetration was observed inside the large interlobular spaces of the pancreas at higher efficiency in the c+dECM as compared to the dECM scaffold (Figure 6A). This difference might be due to the better preservation of the 3D structure of the pancreatic ECM scaffold with the crosslinking step, as compared to a collapsed, thus denser dECM scaffold.

To evaluate the viability of the seeded cells, we performed immunolabelling with Ki67, to visualize cells in the cell cycle, and with Caspase3, to label apoptotic cells. Most of the cell nuclei were Ki67-positive, indicating that these cells were in the cell cycle and alive, most probably in proliferation (Figure 6B). This was observed in both conditions (dECM and c+dECM) and independently of the cell’s position, i.e., distant or in direct contact with the ECM scaffolds. The analysis of cell death showed similar results, with only a few cells being labelled by the anti-Caspase3 antibody and no correlation between these apoptotic cells and their position with respect to the ECM scaffold (Figure 6B). Finally, similar observations were made when pancreatic carcinoma cells PANC-1 were seeded in the ECM scaffolds (Appendix A). These results indicated that cell viability was not affected by the ECM scaffold and both the crosslinking and decellularization steps were not toxic.

## 3. Discussion

Advancing knowledge on the biochemical composition and biomechanical properties of the pancreatic extracellular matrix during development is important not only to improve our understanding of fundamental principles that govern pancreas development in vivo, but also to propose new solutions for tissue engineering or regenerative therapies. A comprehensive analysis of the matrisome of embryonic pancreatic tissue in the mouse has not yet been conducted, and the role of only a very few single components has been investigated [9,10,11,24]. In this study, we provided an extensive characterization of the composition and mechanical properties of the ECM at two developmental stages, E14.5 and P1. Furthermore, we developed a crosslinking/decellularization protocol for the embryonic and postnatal pancreas that preserved matrix 3D organization and is biocompatible with cell survival and expansion.

To accommodate the small size of the embryonic pancreas, we had to adapt and scale down available proteomics processes. We utilized a label-free protocol developed by Ouni et al. [16], but with a reduced amount of initial biological material (only 5 mg). We also introduced an extra fractionation step before mass spectrometry analysis to increase the number of detected proteins. By implementing the final pH reverse-phase fractionation step, we were able to identify over 2000 additional proteins (out of the 5169), including 44 matrisome proteins (out of the 167). The total number of identified and quantifiable proteins (160 matrisome proteins) was comparable, if not higher, than those found in other embryonic or adult species, such as human and porcine pancreas [25,26]. 

The proteomic analysis revealed a division of the samples into two subgroups based on their developmental stage (Appendix A). This division highlighted the distinct proteomic and matrisome profiles of each stage. A comparison of the matrisome distribution revealed a higher detection of collagenous proteins and ECM regulators in the P1 stage, while proteoglycans and secreted factors were more present in the E14.5 stage. This result is compatible with the more mature ECM state in the P1 pancreatic tissue, which is more fibrous (fibrous connective tissue), while the E14.5 ECM is primarily composed of proteoglycans (mucosal connective tissue), and confirmed the observations performed on human pancreatic development [27].

Volcano plot analysis revealed that 52 proteins were more enriched in P1 as compared to only 5 in E14.5 pancreatic tissue. The detection of P1-enriched proteins implicated in islet cell proliferation/regeneration (REG1) and in the digestive function (CELA3A, DMBT1) confirmed the robustness of our analysis [28,29,30]. Note that some ECM proteins were only present at one developmental stage and in two out of three samples, thereby preventing quantification. Five ECM-related proteins were exclusively detected at E14.5, namely SMOC2, PLXNA2, COLEC12, PARM1, and SERPINA10. This most probably reflects the differences between the two developmental stages. Additional experiments targeting specifically these proteins would be required to better understand this observation. One of these, SMOC2, is a protein that enhances the response to angiogenic growth factors, mediates cell adhesion, and has been reported to mediate cell type-specific differentiation during gonad and reproductive tract development [31]. Additionally, the PLXNA2 receptor for the axon-guidance molecule family, SEMAPHORINS, might also deserve further investigation. Indeed, interactomics analysis highlighted that the PLXNA1-SEMA6D pair could be involved in endothelial-to-epithelial communication during pancreas development [32]. Given the potential roles of SMOC2 and PLXNA2, further investigation is needed in the context of pancreatic development. 

The ranking of the most-abundant core matrisome proteins revealed a majority of basement membrane proteins at E14.5, while interstitial proteins were high at P1. Interestingly, we also observed that collagen VI (COL6A1, COL6A2, and COL6A3) was predominant at both stages, followed by the most-ubiquitous collagen I (COL1A1 and COL1A2). Collagen VI is commonly found in connective tissues, associated with basement membranes, and has been demonstrated to support the in vitro viability and survival of human pancreatic islets [33,34]. Proteomics performed on murine adult pancreas revealed that collagen I was the most-abundant collagen followed by collagen VI [26], suggesting a modification in the proportion of collagen VI/I with pancreas maturation, as already reported in porcine pancreatic tissue [35]. Using decellularized pancreas, the authors observed that the proportion of collagen VI decreased from 30% to 21%, while the proportion of collagen I increased from 55% to 69% from young (7 weeks old) to adult (7 months) porcine pancreatic tissue. These findings, along with our results, support a dynamic change in collagen types during development, with collagen VI being more prevalent in embryonic stages and being progressively replaced by collagen I in young and adult pancreatic tissue. However, this collagen shift was not observed in human development, where collagen I is the most-abundant through the entire development (from 18 weeks of gestation to 61 years), but high levels of collagen VI were also detected [25].

As far as the comparison between our results and those on human matrisome is possible (murine E14.5 to human 18 weeks of gestation and murine P1 to human juvenile (5–16 years)), we found some similarities, as well as differences [25]. In both species, we observed a significant increase of LAMA5 and COL4A1 and a decrease of OGN with development. On the contrary, POSTN was more abundant in our perinatal pancreas, while they observed a significant decrease in juvenile pancreas, as compared to fetal stages. These differences (collagen VI and POSTN) could be attributed to the technical procedure with different proteomics setups, to differential development of the fibrous/connective pancreatic capsule (rich in collagen I) in both species or demonstrate the importance of intraspecies variations, which should, therefore, be further investigated or at least considered when proposing engineering solutions.

The mechanical properties of biological tissues are crucial for their function and development. In vitro studies have shown that substrate elasticity influences lineage determination, with a rigid matrix favoring mesenchymal stem cell differentiation towards osteoblasts and a soft matrix favoring neuronal fate regardless of serum conditions [16]. Remodeling of the matrisome content between E14.5 and P1 suggests potential changes in the mechanical properties of the ECM, which could impact cell adhesion, migration, and differentiation. Our observation of increased collagenous proteins at later stages implies a theoretical decrease in porosity and an increase in elasticity [36]. However, our measurements showed a constant elasticity throughout development, regardless of whether the influence of pancreatic cells was included or not. 

One explanation is that any changes in elasticity were minor and lost within the wide range of elasticity measured (200–2000 Pa or 100–500 Pa). Another possibility is that the modification of the matrisome did not alter the mechanical properties of the tissues, yet, and that stiffening likely occurs after birth. Indeed, Goh and colleagues estimated adult pancreas elasticity around 1210  ±  77 Pa, while our measurements in embryos and newborns were about two-fold lower [15]. However, their measurements were performed on 20 µm cryosections of adult pancreatic tissue, while our tissue cryopreservation and sectioning led to a loss of GAG components at E14.5 and P1.Additionally, due to the dense pancreatic parenchyma, their result likely reflects the elasticity of pancreatic acinar cells, rather than the ECM itself. To approximate the elasticity properties of the sole ECM, we used whole or pieces of the pancreas and excluded values similar to those obtained from isolated and cultured P1 pancreatic cells. Therefore, we propose that the embryonic and perinatal pancreas have a relatively soft matrix, with values ranging between 100 Pa and 500 Pa at both stages.

Although AFM has been widely applied for mechanical measurements of biological samples, most existing studies did not investigate viscoelastic properties. However, viscosity is an important factor in tissue development, particularly in cell migration and tissue organization, and most biological tissues exhibit a viscoelastic behavior [19]. Therefore, using a numerical model, we propose a first insight into the viscosity of pancreatic tissue throughout development. A similar distribution of the apparent and unrelaxed elasticity suggests that the pancreatic ECM behaves like an elastic solid. The low apparent viscosity could be attributed to the rearrangement of ECM polymer chains or to the interstitial fluid moving within the embryonic tissue. While we acknowledge that our method for probing and determining the mechanical properties of the pancreas also has its limitations, our results suggest the existence of a soft and low-viscosity ECM in the developing pancreas and recommend the use of biomaterials approaching this rigidity and viscosity.

To improve the determination of ECM elasticity and viscosity, fluorescent probes and transgenic tagged fluorescent mice can be used to visualize the ECM in tissue and precisely place the cantilever. However, our attempts showed that the samples should ideally have a thickness below 50 µm to be able to make the focus on the tip and tissue simultaneously. Cryosections were disregarded due to observed GAG loss, and protocol improvements are needed. An alternative solution involves functionalizing the AFM cantilever tip with antibodies directed against ECM proteins, but this was not considered due to the complex composition of the ECM [37].

To investigate how the identified biochemical and biomechanical parameters of the ECM affect pancreatic development, we developed a novel protocol to generate an ECM scaffold. While decellularization and recellularization protocols have been successfully applied to numerous adult organs from different species [13,38], the decellularization of embryonic tissue for engineering purposes has not been thoroughly explored. In this study, we used a NaOH-based decellularization protocol [23] and found that this process affected the 3D structure of the ECM scaffold. Since this could be due to the non-mature ECM in embryonic tissues, we tried to counteract this loss of shape and architecture by crosslinking the matrix before cell removal. Specifically, we used a non-permeable cellular agent, EDC, which is commonly used to reinforce ECM hydrogels following decellularization, tissue digestion, and lyophilization [39,40]. This approach successfully preserved the 3D structure of the ECM scaffold, as demonstrated by the histological staining and labelling of the basement membrane proteins in 2D and 3D, without preventing cell removal. Although we did not compare the stability of the ECM scaffolds, it is worth noting that crosslinking may improve it. Additionally, the crosslinking followed by decellularization positively impacted (i) sulfated glycosaminoglycan and (ii) cell seeding. 

In the past, NaOH has been used for polysaccharide extraction from tissues by exploiting the sensitivity of the O-glycosylation link to alkali treatment. This sensitivity is probably responsible for the loss of sulfated GAG in our alkaline decellularization process [41]. However, preliminary experiments conducted on the P14 pancreas in two different settings (perfusion or immersion) indicated that increasing the EDC concentration and/or lengthening its incubation time could improve GAG preservation in the ECM scaffold (Appendix A). Indeed, we believe that increasing the time of incubation could allow the formation of covalent links between GAG and proteins of the matrix since EDC has already been used to attach GAG on collagen matrices for tissue engineering [42]. However, modifying these parameters also increases the risk of crosslinking transmembrane proteins in the scaffold. Further optimization is, therefore, necessary to increase GAG contents, while effectively removing cells. This preserved ECM scaffold could then be used to generate better lyophilized biomaterials. 

Seeded cells better penetrated into the c+dECM scaffold, as compared to the dECM scaffold, at least within the large interlobular spaces of the ECM scaffold. However, the seeding conditions clearly need further optimization so that cells effectively penetrate and colonize the intercellular spaces of the decellularized pancreatic ECM. One solution could be to inject cells through the major ducts of the pancreas [15]. However, given the relatively small size of the tissue, a more realistic solution could be to culture the cells and the decellularized tissue in permanent agitation or in a flow to counteract cell sedimentation and to promote cell entry inside the ductal and acinar space of the pancreas [43].

In summary, our work successfully demonstrated the feasibility of detecting ECM proteins in E14.5 and P1 pancreatic tissue, creating one of the largest and first datasets of the murine pancreas proteome and matrisome during development. The power of our approach was demonstrated by the detection of factors only expressed at one developmental stage and that could support pancreatic differentiation/development. While viscoelasticity studies have previously been conducted on mature pancreatic tissues [44], our study presents the first-ever assessment of the embryonic pancreas and its ECM, revealing a relatively soft and low-viscosity matrix during development. We believe that this comprehensive list of proteins and the viscoelasticity properties at the E14.5 and P1 stages are important sources of information that should guide or influence the design of scaffolds for an artificial pancreas, but also help develop protocols for obtaining mature pancreatic cells from embryonic stem cells. Additionally, we developed a novel protocol for generating an ECM scaffold free of cells that retains most of the matrisome proteins. This scaffold can serve as a valuable platform for investigating the matrix effect on pancreatic development. Overall, our findings contribute to a deeper understanding of the role of the ECM in pancreatic development and have potential implications for regenerative medicine. 

## 4. Materials and Methods

### 4.1. Animal Experimentation

Wild-type C57BL/6 mice (Jackson Laboratory, Bar Harbor, ME, USA) were raised and treated according to the NIH Guide for Care and Use of Laboratory Animals. Experiments were approved by the University Animal Ethical Committee, UCLouvain (2016/UCL/MD/005 and 2020/UCL/MD/011), and followed the recommendations of the ARRIVE guidelines. Males and females were mated, and the day of the vaginal plug was considered as Embryonic day (E) 0.5. Pregnant females were sacrificed by cervical dislocation at the desired time point, and embryos at E12.5 and E14.5 were collected. Embryos and Newborns at Postnatal day 1 were dissected under a stereomicroscope, and pancreatic tissue was either directly processed (decellularization and AFM) or flash frozen and stored at −80 °C (mass spectrometry). Postnatal day 14 mice were anesthetized with a xylazine (20 mg/kg)/ketamine (200 mg/kg) solution and sacrificed by cervical dislocation after cardiac flushing with PBS and crosslinking solution. 

### 4.2. Mass Spectrometry (MS)

#### 4.2.1. Sample Preparation for Mass Spectrometry Analysis

The preparation of samples for mass spectrometry analysis followed the protocol described in Ouni et al. [16], with some modifications. Briefly, 5 mg (wet weight) of pancreatic tissue, which corresponded to 20 E14.5 pancreas and 2 P1 pancreas, was lysed mechanically with stainless steel beads (Full Moon Biosystem, LB020, Sunnyvale, CA, USA) and Rapigest homogenization buffer (2% Rapigest (Waters, Milford, MA, USA, 186001861), 0.25 mM sodium orthovanadate, 0.25 mM PMSF, 50 mM NaF, 1 mM cOmplete inhibitor EDTA free (Roche, Basel, Switzerland, 13539320), 25 mM HEPES pH 7.6, and 300 mM NaCl) in Precellys tissue homogenizer (4 × 15 s) at 4 °C. After centrifugation, samples were separated into two fractions: Fraction 1 for the supernatant and Fraction 2 for the pellet. Fraction 2 was digested enzymatically with 1.35 mg/mL Liberase^TM^ DH (Roche, Basel, Switzerland, 492586) for 2 h at 37 °C, resuspended in urea 6 M, and centrifuged at 10,000× *g* for 5 min, and the remaining pellet was discarded. The BCA assay was used to quantify total proteins by subtracting Liberase^TM^ DH amounts for Fraction 2. Then, 200 µg of the proteins from both fractions was reduced with 5 mM DTT for 30 min at 56 °C and alkylated with 25 mM chloroacetamide for 25 min at room temperature. For Fraction 1, the proteins were precipitated using chloroform-methanol phase-extraction, resuspended in 100 mM tetraethylammonium bromide (TEAB), and afterwards, digested with Lys-C/trypsin (Promega, Madison, WI, USA, V507A) at a 1/25 enzyme:substrate ratio (*w*/*w*) overnight at 37 °C. Digestion was stopped with 0.2% trifluoroacetic acid (TFA), and the solution was centrifugated at 16,000× *g* for 10 min at 4 °C. The supernatant was dried in a speedvac and resuspended in 3.5% acetonitrile (ACN) and 0.1% TFA. For Fraction 2, proteins were digested twice, first with Lys-C/Trypsin for 2 h at a 1:50 enzyme:substrate ratio and then with trypsin (Thermo Fischer, Waltham, MA, USA, 90057) overnight at a 1:100 enzyme:substrate ratio, interspersed with a reduction of the urea concentration to 1M using 100 mM TEAB. After stopping the digestion with TFA 0.5%, the solution was centrifugated at 16,000× *g* for 10 min. Then, the peptides were desalted and concentrated using HyperSep C18 Cartridge 200 mg/mL (Thermo Fischer, Waltham, MA, USA, 60108-303). After drying in a speedvac and similarly to Fraction 1, the peptides were resuspended in 3.5% ACN and 0.1% TFA. After the peptide quantification with a Pierce quantitative colorimetric Peptide assay (Thermo Fischer, Waltham, MA, USA, 23290), 100 µg of both fractions were eluted into six fractions using a Pierce High pH Reversed-Phase Peptide fractionation kit (Thermo Fischer, Waltham, MA, USA, 84868) according to the manufacturer’s instruction. Finally, all the fractions were quantified using the Pierce quantitative colorimetric Peptide assay and resuspended in 3.5% ACN and 0.1% TFA for MS injection.

#### 4.2.2. Liquid Chromatography-Tandem-Mass-Spectrometry 

One microgram of peptides dissolved in Solvent A (0.1% TFA in 2% ACN) was directly loaded onto a reverse-phase pre-column (Acclaim PepMap 100, Thermo Scientific, Waltham, MA, USA) and eluted in backflush mode. Peptide separation was achieved using a reverse-phase analytical column (Acclaim PepMap RSLC, 0.075 mm × 250 mm, Thermo Scientific, Waltham, MA, USA) with a 90 min linear gradient of 4–27.5% Solvent B (0.1% TFA in 80% ACN) for 37 min, 27.5–50% Solvent B for 20 min, 50–95% for 10 min, and holding at 95% for the last 10 min at a constant flow rate of 300 nL/min on an Ultimate 3000 RSLN nanoHPLC system (Thermo Scientific, Waltham, MA, USA). The peptides were analyzed by an Orbitrap Fusion Lumos tribrid mass spectrometer (Thermo Fisher Scientific, Waltham, MA, USA) with enabled advanced peak determination (APD). The peptides were subjected to a nanospray ionization source, followed by MS/MS in the Fusion Lumos coupled online to the HPLC. Intact peptides were detected in the Orbitrap at a resolution of 120,000, and MS/MS spectra were acquired in the IT after HCD fragmentation at 30%. A data-dependent procedure of MS/MS scans was applied for the top precursor ions above a threshold ion the count of 5.0 × 10^3^ in the MS survey scan with 40 s dynamic exclusion. The total cycle time was set to 4 s. MS1 spectra were obtained with an AGC target of 4 × 10^5^ ions and a maximum injection time of 50 ms. MS2 spectra were acquired with an AGC target of 1 × 10^4^ ions and a maximum injection time of 35 ms. For MS scans, the *m*/*z* scan range was 350 to 1800. MS/MS spectra were exported using the following settings: peptide mass range: 350–5000 Da; minimal total ion intensity: 500.

#### 4.2.3. Protein Identification and Quantification

The resulting MS/MS data were processed using the Sequest HT search engine within Proteome Discoverer 2.5 against a mouse protein reference target–decoy database obtained from Uniprot (1 January 2022, 55,310 forward entries). Trypsin was specified as the cleavage enzyme, allowing up to 2 missed cleavages, 4 modifications per peptide, and up to 3 charges. The mass error was set to 10 ppm for precursor ions and 0.1 Da for fragment ions, and the considered dynamic modifications were +15.99 Da for oxidized methionine proline and +57.00 Da for carbamidomethyl cysteine. Label-free quantification (LFQ) was performed by measuring the area under the curve (AUC) for each peptide, and the abundance ratios were calculated within Proteome Discoverer 2.5 after normalization on the total ionic current (TIC). The false discovery rate (FDR) was investigated using Percolator, and the thresholds for proteins, peptides, and modification sites were specified at 1%.

#### 4.2.4. Data Processing

ECM-related genes in the raw MS data were identified and annotated using a murine in silico-identified ECM dataset from the Matrisome Project. The proteins were then filtered out to keep the most-confidently detected proteins for subsequent interpretation of the results. This threshold included proteins with unique peptides equal to or higher than 2, a false discovery rate of less than 1%, and a minimum of 3 peptide spectral matches. If found in both fractions, protein abundance was defined as the addition of its abundance in Fraction 1 and Fraction 2. Once all the samples were combined (n = 3 for E14.5 and n = 3 for P1), proteins that were only detected in one or two samples were removed, and the median absolute values method was applied to normalize the datasets. 

#### 4.2.5. Statistical Tests

All statistical tests on proteomics data were performed using the R software (v.3.5.1, Bruxelles, Belgium). Principal component analysis was computed on all quantifiable proteins before normalization using the Nipals package, which considers the influence of the NA values. Since a batch effect was observed between our samples, we decided to perform a paired *t*-test to compare the proteins between the E14.5 and P1 samples. This was computed with the Lima package with P1−E14.5 as a fold change and considering the batch effect not similar for proteins of the same dataset. The proteins were considered significantly different when the adjusted *p*-value was under 0.05. The listing of the most-abundant proteins was performed based on the number of peptide spectral matches at both stages and compared with a paired *t*-test. Both analyses are available in the Appendix A. 

### 4.3. Atomic Force Microscopy

#### 4.3.1. Preparation of Samples for Atomic Force Microscopy

Pregnant or newborn mice were sacrificed two hours before the AFM analysis. Pancreatic tissue was dissected under a stereomicroscope, cut into small pieces (P1), and deposited in AFM dishes (WillCo dish, Amsterdam, The Netherlands, GLOST-3522) previously coated with Poly-L-Lysine 0.01% (Sigma, St. Louis, MO, USA, P4707) or polyethyleneimine 4 mg/mL (Sigma, St. Louis, MO, USA, 408727-100ML). The tissues were kept in DMEM medium containing 25 mM HEPES (Gibco, New York, NY, USA, 21063-029) + 10% FBS (Lonza, Basel, Switzerland, DE14-850F) + 1% Pen/Strep (Gibco, New York, NY, USA, 15140-122) in an incubator at 37 °C and 5% CO_2_ at least 1 h before AFM analysis.

To isolate pancreatic cells, the P1 pancreas was incubated with collagenase (1.4 mg/mL Collagenase P (Sigma, St. Louis, MO, USA, 11213865001), DMEM serum-free calcium (Gibco, New York, NY, USA, 21068-028) + 1% Pen/Strep and 20 U/mL DNase I (Sigma, St. Louis, MO, USA, D4527-40KU)) for 30 min at 37 °C, under constant shaking (700 rpm). Digestion was stopped by adding an inactivation solution (DMEM serum-free calcium + 1 mM EDTA + 20% FBS + 1% Pen/Strep), and the digested mixture was filtered with a 40 µm cell strainer (Corning, New York, NY, USA, 431750). The filtrate was then centrifugated at 600× *g* for 4 min, and the pellet was resuspended in classic DMEM + 10% FBS + 1% Pen/Strep and plated in AFM dishes overnight in an incubator. The AFM measurements were performed after 36 h on confluent cells and the medium changed to the one utilized for pancreatic tissues.

#### 4.3.2. Atomic Force Microscopy

All AFM measurements were performed under culture conditions, at 37 °C and in the same medium as described before. The AFM JPK Nanowizard 3 software was used to conduct the experiments. AFM tips for probing pancreatic tissue were obtained by gluing a 6 µm in diameter silica microsphere on NP-010 probes with nominal spring constants of 0.06 N/m. Spherical beads were used to minimize damage and calculate the global rigidity of the tissue. Cantilevers were first calibrated using the thermal noise method, yielding values ranging from 0.06 to 0.09 N/m. The samples were scanned in force–volume mode using a constant approach speed of 10 µm/s, a set point force of 5 nN, and a retraction length of 30–50 µm. For the P1 pancreatic cells, AFM tips, and 1940 nm-diameter spherical beads on MLCT-SPH-1µm-DC probes with nominal spring constants of 0.01 N/m were purchased from Bruker, Billerica, MA, USA. The cells were scanned in force–volume mode using a constant approach speed of 1 µm/s, a set point force of 5 nN, and a retraction length of 5 µm. Several force–volume maps of dimensions 100 µm × 100 µm with 16 × 16 pixels and 10 µm × 10 µm with 8 × 8 pixels were recorded on tissue samples and on cells, respectively.

#### 4.3.3. Data Analysis

The AFM images and FD curves were analyzed using JPK SPM data Processing Version 6.1.172. A Hertzian model, with a spherical tip shape and a Poisson ratio of 0.5, was fit to the approach part of the force–indentation curves to quantify the apparent Young’s modulus (E apparent). To remove the effects of proteins and debris floating on the tissue surface, the curves were fit after the contact point, from 1 to 5nN. After confirming the Gaussian distribution for each biological sample (n = 7 at E12.5, n = 7 at E14.5, and n = 3 at P1), the values of Young’s modulus were pooled together for each stage. Outliers of the Young’s modulus values at each stage were detected using the ROUT method (Q = 0.1%) and removed.

#### 4.3.4. Kelvin–Voight–Maxwell Model

Each force–displacement curve was imported and analyzed in Pyjibe using the Kelvin–Voight–Maxwell (KVM) model developed by Abuhattum et al. [22]. The contact point was estimated using the piecewise fit with the line and polynomial, and the Hertz model corrected for viscoelasticity using the KVM model was applied to fit the approach curve. For each curve, four parameters (E unrelaxed, E apparent, Maxwell relaxation, and viscosity) and a fit rating (from 0 to 10) were extracted. The curves with a score below 4.5 (or 3.5 for E14.5) were filtered out, and the results were pooled by developmental stages. Outliers were detected as described before and removed if present in any of the four parameters. To assess the difference in the viscoelastic properties, an unpaired *t*-test (*p* < 0.05) was performed between the E12.5 and E14.5 or E14.5 and P1 pancreatic tissue.

### 4.4. ECM Scaffolds

#### 4.4.1. Crosslinking and Decellularization

Harvested pancreatic tissues were directly processed after dissection or thawed slowly (for E14.5) to room temperature before starting the crosslinking and decellularization process for the generation of pancreatic ECM scaffolds. First, the samples were incubated in crosslinking solution (5 mM 1-ethyl-3[3-dimethylaminopropyl) carbodiimide hydrochloride (EDC) (Thermo Scientific, Waltham, MA, USA, PG82074), 0.1 M MES buffer pH 4.7 (Sigma, St. Louis, MO, USA, M-8250)) at room temperature and under active agitation during 5 min in order to form amide bonds between carboxylic acid and the primary amine group. The samples were then rinsed with Milli-Q water before beginning the decellularization process. Based on the protocol described by van Steenberghe et al. [23], the samples were successively incubated in acetone for 30 min (E14.5) or 2 h (P1) with a change of solution at mid-time, in 0.01 M NaOH (Fischer Chemical, Hampton, NH, USA, J/7620115) until the tissues became entirely transparent (about 8 min for the E14.5 tissue and 20 min for P1) and in 500 U DNase I solution at 37 °C during 30 min (E14.5) or 2 h (P1). All incubation steps were performed on an agitator plate and interspersed with extensive Milli-Q water rinsing at room temperature. 

Crosslinking of P14 pancreatic tissues was achieved using two protocols. The classical one consisted of the incubation of micro-dissected pancreas with different EDC concentrations (5 and 20 mM) and increasing periods of time (5, 20, and 40 min) in the same conditions as described above. In the second method, anaesthetized mice were perfused with PBS for 1 min and then with different EDC concentrations (5, 15, 30, 60, or 90 mM) for 3 min via the left ventricle. The pancreas was thereafter dissected, and the decellularization protocol continued as described above using a 0.1 M NaOH solution.

For histological staining, immuno-fluorescence, and immuno-histochemistry labelling, the ECM scaffolds were fixed in 4% paraformaldehyde overnight at 4 °C and embedded in paraffin using a Tissue-Tek VIP-6 (Sakura), and sections of 7 µm were prepared. For recellularization, the ECM scaffolds were rinsed with PBS and 1% Pen/Strep for 3 days.

#### 4.4.2. GAG Quantification

After completion of the crosslinking and decellularization process, three P1 scaffolds or one P14 ECM scaffold were flash frozen and lyophilized. The powder was solubilized and digested overnight at 65 °C (0.2 M Na_2_HPO_4_/NaH_2_PO_4_ buffer (pH 6.4), 0.01 M EDTA disodium salt dihydrate, 0.14 mg/mL papain (Sigma, St. Louis, MO, USA, P3125), 0.09 M sodium acetate (Merck, Rahway, NJ, USA, 6268) and 5 mM cysteine HCl (Sigma, St. Louis, MO, USA, 30120)) under gentle shaking. After centrifugation, 100 µL of supernatant was collected for the quantification of sulfated glycosaminoglycans using the Blyscan sGAG assay (Biocolor, Carrickfergus, Northern Ireland, B1000) according to the manufacturer’s instruction [45].

#### 4.4.3. Cell Culture

Embryonic mouse stem cells (E14tg2A), obtained from BayGenomics, were cultured with a daily change of medium (GMEM (Sigma, St. Louis, MO, USA, G5154) + 2 mM GlutaMax (Sigma, St. Louis, MO, USA, 35050-038) + 1 mM sodium pyruvate (Gibco, New York, NY, USA, 11360-070) + 1X Non-essential amino acid (Gibco, New York, NY, USA, 11140-050) + 10% FBS + 1% Pen/Strep + 100 µM beta-mercaptoethanol (Thermo Fischer, Waltham, MA, USA, 31350-010) + 10 ng/nL recombinant mouse LIF protein (R&D, Minneapolis, MN, USA, 8878-LF-025/CF)) in plastic dishes coated with 0.1% gelatin. The pancreatic carcinoma cell line PANC-1 (ATCC, Manassas, VA, USA, CRL-1469) was cultured in DMEM (Gibco, New York, NY, USA, 31966-021) + 10% FBS + 1% Pen/Strep with a change of medium every three days. All cells were passaged at least twice before starting the recellularization process.

#### 4.4.4. Recellularization

The ECM scaffolds were incubated in cell culture medium for 4 h before seeding. Afterwards, tissues were placed on microporous membranes (Millipore, Burlington, MA, USA, PICM01250) with 330 µL of medium underneath. One-million E14tg2A or PANC-1 cells were seeded on top of the decellularized tissue. Seeded scaffolds were maintained in culture for 6 days with daily medium change. The culture was stopped after 2, 4, and 6 days, and the recellularized tissues were fixed for 2 h with PFA 4% at 4 °C, followed by equilibration in PBS-20% sucrose solution and embedding in PBS-15% sucrose-7.5% gelatin, and sections of 7 µm were prepared. 

#### 4.4.5. Histological Staining and Immunohistochemistry of Mouse Tissue Sections

Prior to histological analysis, paraffin sections were deparaffinized with xylene 2 min × 5 min and gradually rehydrated, while gelatin sections were immersed in 40 °C PBS for 10 min. For histological staining, sections were immersed in hematoxylin and eosin to assess the presence of cellular components, Sirius red/fast green for the preservation of collagenous proteins, and Alcian blue pH 2.5 (Vector Laboratories, Mowry Ave Newark, CA, USA, H-3501) to visualize GAG components. 

For the detection of hyaluronic acid in tissue sections, sections were immersed in an H_2_O_2_ solution for 30 min to inhibit endogenous peroxidases. Sections were then immersed in citrate buffer (0.01 M citric acid and 0.05% Tween 20 (pH 6.0)) for antigen retrieval, permeabilized 3 min (PBS/0.3%Triton X-100), blocked for 45 min (PBS/0.3% Triton X-100/10% BSA/3% milk), and incubated with hyaluronic acid binding protein biotinylated (dilution 1/100, Merck, Kenilworth, NJ, USA, 385911) overnight at 4 °C. After 3 × 3 min of rinsing, sections were incubated with streptavidine POD (PBS/0.3% Triton X-100/10% BSA) for 1 h at room temperature. After a final 3 × 3 min rinsing, sections were immersed in DAB solution (1/1000, Abcam, Cambridge, United-Kingdom, ab64238) for 30 s, followed by hematoxylin for 3 min and tap water for 5min.

All slides were mounted in Dako aqueous Medium (Agilent Technologies, Santa Clara, CA, USA, S3025) and scanned with the panoramic P250 digital slide scanner (objective 20×).

#### 4.4.6. Immunofluorescence Labelling on Wholemount and Sections of Mouse Tissue

The pancreatic tissues and ECM scaffolds were labelled and imaged as previously described in Glorieux et al. [46]. For immunofluorescence labelling of mouse tissue sections, sections were incubated with primary antibodies at the appropriate dilution (1/200 for LAMININ (Sigma, St. Louis, MO, USA, L9393) and COLLAGEN IV (Millipore, Burlington, MA, USA, ab756P), 1/250 for E-CADHERIN (Cell Signaling, Danvers, MA, USA, 3195) and KI67 (BD Pharmingen, San Diego, CA, USA, 556003), and 1/100 for cleaved CASPASE 3 (Cell Signaling, Danvers, MA, USA, 9661)). Images were acquired with a Cell Observer Spinning Disk Confocal Microscope (Zeiss, Oberkochen, Germany) using a 25× or 100× immersive oil objective.

For wholemount, P1 pancreatic tissues and ECM scaffolds were first incubated with LAMININ or COLLAGEN IV at a dilution of 1/100 and, afterwards, with secondary antibodies and HOECHST nuclear counterstain at a dilution of 1/500. Samples were either embedded in agarose 1% and imaged with LSM980-Multiphotons (Zeiss, Oberkochen, Germany) and a 25× objective or cleared for five weeks using Cubic solutions (two weeks in Cubic1 and three weeks in Cubic2) [47], glued to a needle, and imaged by a light-sheet microscope (Z.1, Zeiss, Oberkochen, Germany) using a 20× objective.

## 5. Conclusions

In this study, we carried out a comprehensive examination of the proteome and determined the mechanical properties of the pancreas during development, at E14.5 and in newborn, at P1. We successfully applied a tailored proteomic workflow to identify and quantify matrisome proteins and measured the viscoelasticity properties of the micro-dissected pancreas using an advanced AFM technique. Additionally, we proposed a novel protocol (crosslinking + decellularization) for generating ECM scaffolds with preserved 3D organization, an opportunity window for GAG maintenance and cell biocompatibility. These findings hold great potential for tissue engineering applications and the optimization of cell differentiation protocols, providing valuable insights and methodologies for researchers in the field. 

## Figures and Tables

**Figure 1 ijms-24-10268-f001:**
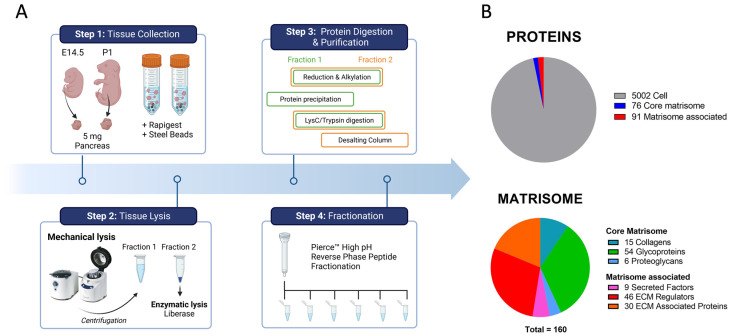
Workflow for in-depth proteome-wide quantification of pancreatic extracellular matrix (ECM) at embryonic (E) day 14.5 and postnatal (P) day 1 stages. (**A**) To identify proteins from pancreatic tissues, we used a multi-step sample preparation protocol. In Step 1, 5 mg wet weight of pancreatic tissues collected at E14.5 and P1 was mixed with stainless steel beads and MS-compatible reagents. In Step 2, the samples were subjected to a first mechanical lysis using a tissue homogenizer. The resulting mixture was then separated into two fractions: the supernatant (Fraction 1) and the remaining pellet (Fraction 2). Fraction 2 was further lysed enzymatically to release entangled proteins of the core matrisome and solubilized in urea. In Step 3, both fractions were subjected to protein reduction and alkylation. For Fraction 1, proteins were first precipitated and then digested into peptides with Lys-C/Trypsin, while for Fraction 2, proteins were first digested and the peptides further purified using a C18 desalting column. In Step 4, a reverse-phase high-pH fractionation step was performed to decomplexify the samples and increase the detection of less-abundant proteins. The eluted fractions were then dehydrated, resuspended in MS sample solvent, and finally, injected into the mass spectrometer for protein identification with label-free quantification. Figure created with BioRender.com. (**B**) Proteins and matrisome detected in E14.5 (n = 3) and P1 (n = 3) pancreatic tissue by liquid chromatography coupled with tandem mass spectrometry (LC-MS/MS). In total, 5169 proteins were confidently detected (unique peptides ≥ 2 and peptide spectral matches ≥ 3) including 5002 proteins related to the cellular compartment (grey), 76 to the core matrisome (blue), and 91 associated with the matrisome (red). After keeping proteins detected in at least three experiments (out of 6), the matrisome was further categorized with collagens (turquoise), ECM glycoproteins (green), and proteoglycans (blue) for the core matrisome and secreted factors (pink), ECM regulators (red), and ECM-affiliated proteins (orange) associated with the matrisome.

**Figure 2 ijms-24-10268-f002:**
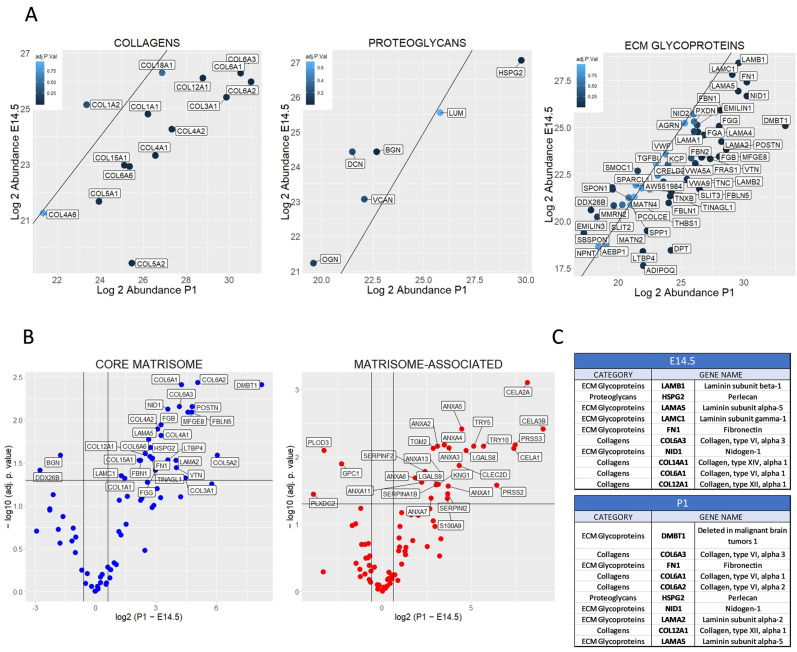
Differential matrisome profile between E 14.5 and P1 pancreatic tissue. (**A**) Scattered plots of the log2 abundance of matrisome proteins in E14.5 pancreas (*y*-axis) vs. abundance in P1 pancreas (*x*-axis). The left plot displays collagens, the middle plot proteoglycans, and the right plot ECM glycoproteins. The straight black line represents the median of the graph, with proteins detected more in E14.5 pancreas (above the line) and proteins detected more in P1 pancreas (below the line). Colors indicate the adj. *p*. value of the paired *t*-test performed between proteins of E14.5 (n = 3) and P1 (n = 3). (**B**) Volcano plots of core matrisome (blue) and matrisome-associated (red) proteins based on the difference between P1 and E14.5 (P1−E14.5). Significant proteins (adj. *p* value < 0.05, difference 1.5) are labeled in both plots. (**C**) List of the ten most-abundant core matrisome proteins in E14.5 and P1 pancreatic tissue.

**Figure 3 ijms-24-10268-f003:**
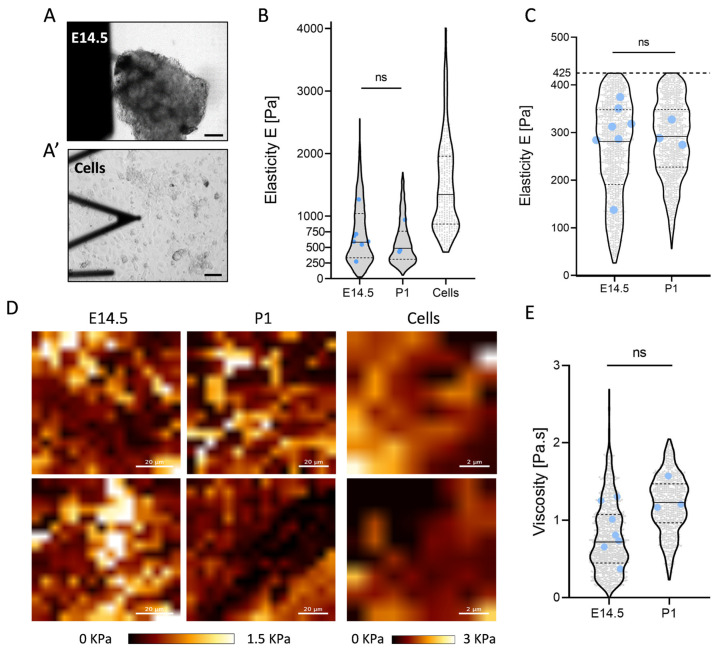
Similar viscoelasticity properties of pancreatic tissue and ECM during development. (**A**,**A’**) Optical images of E14.5 pancreatic tissue (**A**) and pancreatic cells (**A’**) probed by atomic force microscopy (AFM) cantilever. Top image: scale = 100 µm; bottom image: scale = 60 µm. (**B**) Global elasticity (Pa) of E14.5 and P1 pancreatic tissues, as well as P1 pancreatic cells. Grey points represent individual elasticity value computed from one force–displacement curve. Blue points represent the mean of elasticity for one tissue sample (n = 7 at E14.5, n = 3 at P1). (**C**) Approximate elasticity of pancreatic ECM at E14.5 and P1. Values within the range of cell’s elasticity computed in (**B**) are excluded in E14.5 and P1 pancreas to display hypothetical elasticity related to the ECM. (**D**) Examples of elasticity maps at two distinct locations for E14.5, P1 pancreatic tissue, and P1 pancreatic cells. For both tissues, maps were obtained from an analyzed area of 100 µm × 100 µm with 32 acquisitions per side. For the cells, maps were obtained from an analyzed area of 10 µm × 10 µm with 8 acquisitions per side. The colored squares indicate the values of elasticity from low (0 KPa, dark) to high (1.5 for tissues or 2 KPa for cells, bright) calculated from the force–displacement curve of the AFM cantilever. Left and middle panels: scale = 20 µm; right panel: scale = 2 µm. (**E**) Viscosity (Pa.s) of E14.5 (n = 7) and P1 (n = 3) ECM based on the numerical model developed by Abuhattum et al. [22]. Unpaired *t*-test performed on the mean of biological replicates (blue dots) for each graph. ns = non-significant.

**Figure 4 ijms-24-10268-f004:**
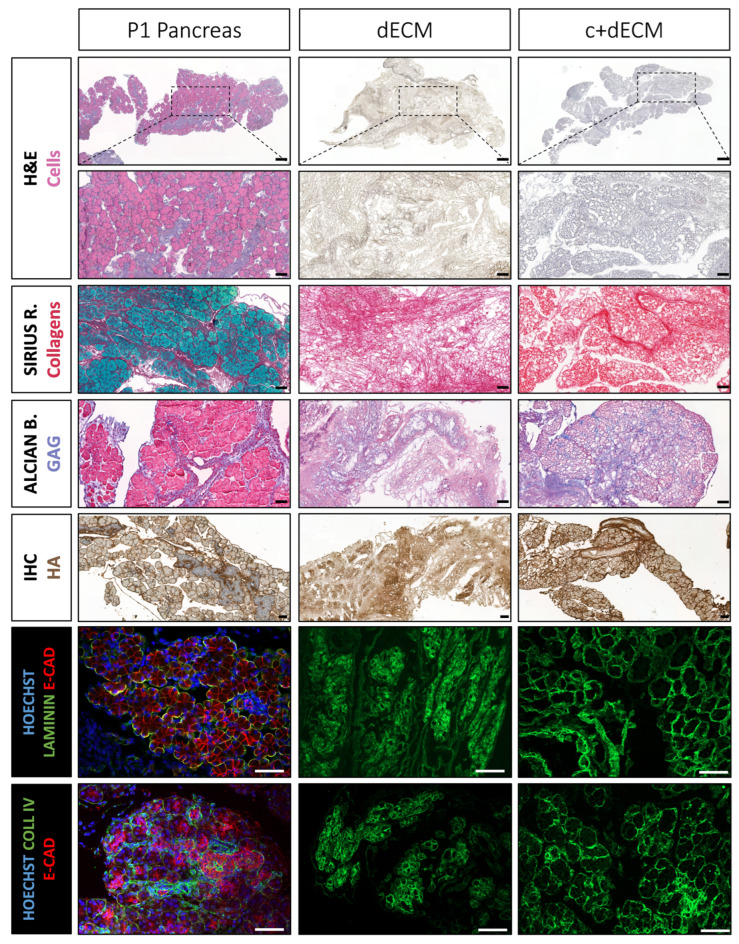
Better preservation of decellularized pancreatic ECM after crosslinking. The morphology of the P1 pancreatic ECM was evaluated in three conditions: native tissue (left, P1 pancreas), NaOH-based decellularized pancreatic (middle, dECM), and EDC-crosslinked and NaOH-based decellularized pancreatic (right, c+dECM). Tissues were stained with H&E to assess the removal of cells at low (first line) and high magnification of the selected area (black dotted rectangle, second line), Sirius red to evaluate the preservation of the collagenous structure (third line), and Alcian blue for the preservation of glycosaminoglycans (fourth line). Sirius red staining was counterstained with fast green and Alcian blue with eosin for the detection of cytoplasmic content. Immunohistochemistry (IHC) and immunofluorescence labelling were performed with biotinylated hyaluronic acid (HA) binding protein (fifth line) and antibodies directed against E-CADHERIN (red), LAMININ (green, sixth line), or COLLAGEN IV (green, seventh line). HA labelling was counterstained with hemalun, while immunofluorescence labelling were counterstained with the nuclear stain HOECHST (blue). Scale bars: 200 µm (line 1), 50 µm (line 2–7).

**Figure 5 ijms-24-10268-f005:**
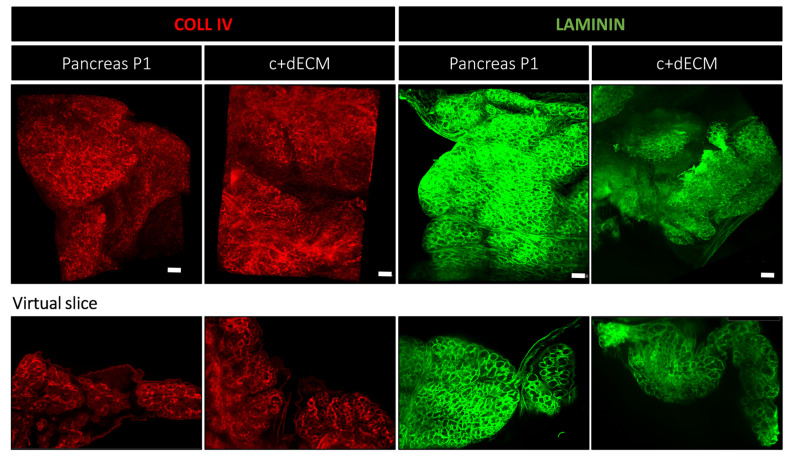
Overall preservation of 3D structure in c+dECM scaffold. Whole native pancreatic tissue and c+dECM scaffolds were labelled with antibodies directed against COLLAGEN IV (red) and LAMININ (green). A similar macrostructure was observed in both conditions, demonstrating the efficiency of EDC to crosslink ECM proteins and preserve the 3D structure. Three-dimensional images were obtained with light sheet microscopy after clearing in Cubic solution. The bottom images represent a virtual slice from the above 3D objects. Scale bar: 300 µm.

**Figure 6 ijms-24-10268-f006:**
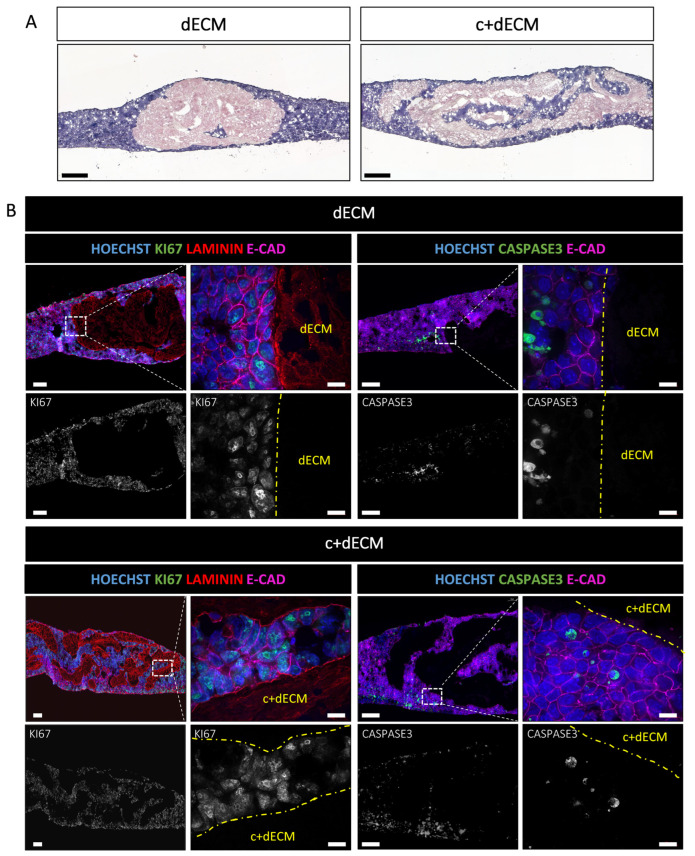
Biocompatibility of dECM and c+dECM scaffolds with embryonic stem cells. dECM and c+dECM scaffolds were seeded with one-million embryonic stem cells and cultured for 6 days. (**A**) H&E staining of the recellularized ECM scaffolds showed the aggregation of cells around the scaffolds for both conditions. A slightly higher penetration was observed for c+dECM with cells located inside interlobular spaces. (**B**) Immunolabelling of the recellularized ECM scaffolds with antibodies directed against KI67 (green) or CASPASE 3 (green), E-CADHERIN (purple), and LAMININ (red) and counterstained with HOECHST (blue) demonstrated the biocompatibility of both decellularization processes. Top panels include a low-magnification image of the entire structure (**left**) and a zoomed-in image (white dotted rectangle) for cellular resolution (**right**). The bottom panels show the KI67 or CASPASE 3 single channel (white) from the corresponding top images. Scale bars: 200 µm (**A**), 100 µm (**B**, **left** images) and 10 µm (**B**, **right** images).

## Data Availability

The mass spectrometry proteomics data have been deposited into the ProteomeXchange Consortium via the PRIDE [1] partner repository with the dataset identifiers PXD042423 and 10.6019/PXD042423.

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
