# Peer review of "In-Depth Analysis of the Pancreatic Extracellular Matrix during Development for Next-Generation Tissue Engineering"

_ijms, 2023, doi:10.3390/ijms241210268_

Round 1

Reviewer 1 Report

The current work focuses on the In-depth analysis of the pancreatic extracellular matrix during
development for next generation tissue engineering
. The experimental work appears to have been carried out well. However, a few points deserve attention for further publication. I suggest that it is accepted for publication after the following revisions:

- ABSTRACT: In-depth analysis of the pancreatic extracellular matrix during
development for next generation tissue engineering
. What parameters were optimized? Authors must include numbers with the results found. Stability of matrix? How much matrix were utilized to process? Furthermore, what are the conditions of reactions? Temperature, pH, ionic strength, for example. This information should be included in the abstract.

- INTRODUCTION:

- In this study, In-depth analysis of the pancreatic extracellular matrix during
development for next generation tissue engineering
: physical and covalent were implied for preparation? What the advantages? Additionally, the spacer arm, the steric hindrances of matrix reaction caused by this groups when compared to the others groups? These strategies used should be better explained in the manuscript.

- pancreatic extracellular matrix are very special, having a peculiar mechanism of action. This information must be clear in the introduction to present manuscript.

- A paragraph describing the properties, application, mechanism of actuation to pancreatic extracellular matrix must be included in the manuscript.

- What is the origin of this pancreatic extracellular matrix? Is it a commercial ? Modified? How was it produced?

- pancreatic extracellular matrix: Improved kinetic, and efficient increased the activity? This process needs to be explained in the introduction of the manuscript.

- analysis of the pancreatic extracellular matrix during
development for next generation tissue engineering
: What optimization strategy was used? Why was it used? This information needs to be explained in the introduction of the manuscript.

- the pancreatic extracellular matrix during
development for next generation tissue engineering
 presented were compared with a commercial material? This information must be clear in the introduction.

- The contribution and importance of these studies in the work performed must be explained in the introduction of the manuscript.

MATERIALS:

- Include the concentration of solutions.

METHODS:

- Include the molar concentration of all the chemicals used, the way the methods are presented, not possible reproducibility.

- the pancreatic extracellular matrix during
development for next generation tissue engineering
 : Please include more details, temperature, pH, molar ratio, ionic strength.

- RESULTS AND DISCUSSION:

- The influence of substrate systems to  pancreatic extracellular matrix during
development for next generation tissue engineering
  showed how about stability?

- The thermal stability to pancreatic extracellular matrix prepared is one of the most important application criteria for diferent applications. This stability depends to pancreatic extracellular matrix preparation strategy. It also depends on the stabilization of the pancreatic extracellular matrix. This discussion could be improved. Please include in the manuscript.

- The stability in organic solvents, metal ions, or detergent enables its wide application in synthesis processes which nowadays are in great demand from the point of view of industrial. The effect of organic solvents on the pancreatic extracellular matrix activity was studied? For example, in the presence of ethanol, methanol, dimethyl sulfoxide (DMSO), dioxane, n-hexane, tert-butanol, acetone or 2-propanol?

- Was determined the full loading of pancreatic extracellular matrix prepared under the optimized conditions? This information must be clear in the manuscript.

- The pancreatic extracellular matrix may experience protein aggregation (mainly near to the isoelectric point). This may be caused by undesired pancreatic extracellular matrix - interactions where inactivation that can stabilize incorrect pancreatic extracellular matrix structures.

- The optimization of pancreatic extracellular matrix preparation process, the preparations shown having diffusion limitations? Considering the strategy presented in this manuscript. Please, this should be explained in the manuscript. What were the optimum conditions?

- Effect of solution pH since the solution pH affects the generation of hydroxyl radicals and also influences the surfasse charge and interface potential properties of the catalyst, it is one of the important factors. pancreatic extracellular matrix showed considerable improves in the kinetic parameters in terms of activity, specific activity, Km and Vmax, optimum pH and Temperature?

- Reusability of pancreatic extracellular matrix:  The reusability of pancreatic extracellular matrix particles is essential while considering reactions. pancreatic extracellular matrix reusability was accounted for? Reusability studies showed that the remaining pancreatic extracellular matrix assay was obtained to reduce with the increasing number of re-use cycles. The reusability of pancreatic extracellular matrix without alteration in its load capacity of performance with the resulting is an advantage. After cycles, please, explain these results. What other factors can influence the results achieved? In addition, the results should be compared with other works of literature in the same application line.

- Enhanced stability of pancreatic extracellular matrix prepared: Other factors that cause the loss of durability and stability should be explained in the manuscript.

- Please, check all references according to the author's instructions.

- Include more details in the figures (error bars) and tables captions.

- The manuscript must be formatted according to the journal's standards.

Minor editing of English language required

Reviewer 2 Report

The research manuscript entitled “In-depth analysis of the pancreatic extracellular matrix during development for next generation tissue engineering” authored by Christophe E. Pierreux et al. presents a systematic and structured compilation of experimental results from in-depth analysis of the composition and mechanical properties of the ECM during development, with a focus on two key stages: E14.5, and perinatal stage 1 (P1).

 1.      Authors are requested to add a description about novelty of this research and its significance which are absent in previously reported research documents.

 2.       Lines “Although much is known about the intrinsic factors that control pancreas development, understanding of the microenvironment surrounding pancreatic cells remains incomplete.” What are the lacunae in the understanding existing? The authors can explain in a brief paragraph.

3.       What is the motivation behind selecting the study methods using “mass spectrometry” and atomic force microscopy”?

4.       The studies to identify and quantify the ECM composition of the developing pancreas at embryonic E14.5 and perinatal P1 stages, and to measure the biomechanical properties of the ECM which resulted in finding the whole pancreas ECM is relatively soft with no significant change during pancreas maturation. Authors did not consider the microrheological measurement technique to assess the biomechanical properties instead. Why?

 5.       Our findings provide insights into the composition and biomechanics of the pancreatic embryonic and perinatal ECM, offering a foundation for future studies investigating the dynamic interactions between the ECM and pancreatic cells; Authors can explain shortcomings of this research study and future directions of this study.

 6.      Lines 230-235: Our results showed that the elasticity of pancreas ranged from 100 to 2000 Pa, with no significant difference between the two stages (Figure 3B). When probing fresh pancreatic tissues, we assumed that we mostly encountered the extracellular matrix of the micro-dissected tissue. However, we probably also encountered surrounding pancreatic mesenchymal cells or fibroblasts, as suggested by the non-uniform distribution of elasticity in maps at both stages (Figure 3D). The images in Figure 3D are not clear. Authors are requested to provide clean and clear AFM images to explain the context appropriately.

 7.      The indentation experiments showed no noticeable difference in the biomechanical properties of the two selected stages. Henceforth, do the authors recommend ruling out this study in future research? If so, why?

 8.      Authors are requested to consult the following references. There are plenty of references on this subject area available. The claim “Lines 552-553: we provide the first-ever assessment of the viscoelasticity properties of the embryonic pancreas and its ECM, demonstrating a relative soft and low viscous matrix during development” can be justified appropriately.

References https://pubmed.ncbi.nlm.nih.gov/32984301/

https://www.sciencedirect.com/science/article/abs/pii/S1742706117307304

https://faseb.onlinelibrary.wiley.com/doi/abs/10.1096/fj.202200807R

 9.      The conclusions of the study need to be rephrased as it looks very weak.

10.  I recommend a minor revision of the manuscript.

The research manuscript entitled “In-depth analysis of the pancreatic extracellular matrix during development for next generation tissue engineering” authored by Christophe E. Pierreux et al. presents a systematic and structured compilation of experimental results from in-depth analysis of the composition and mechanical properties of the ECM during development, with a focus on two key stages: E14.5, and perinatal stage 1 (P1).

 1.      Authors are requested to add a description about novelty of this research and its significance which are absent in previously reported research documents.

 2.       Lines “Although much is known about the intrinsic factors that control pancreas development, understanding of the microenvironment surrounding pancreatic cells remains incomplete.” What are the lacunae in the understanding existing? The authors can explain in a brief paragraph.

3.       What is the motivation behind selecting the study methods using “mass spectrometry” and atomic force microscopy”?

4.       The studies to identify and quantify the ECM composition of the developing pancreas at embryonic E14.5 and perinatal P1 stages, and to measure the biomechanical properties of the ECM which resulted in finding the whole pancreas ECM is relatively soft with no significant change during pancreas maturation. Authors did not consider the microrheological measurement technique to assess the biomechanical properties instead. Why?

 5.       Our findings provide insights into the composition and biomechanics of the pancreatic embryonic and perinatal ECM, offering a foundation for future studies investigating the dynamic interactions between the ECM and pancreatic cells; Authors can explain shortcomings of this research study and future directions of this study.

 6.      Lines 230-235: Our results showed that the elasticity of pancreas ranged from 100 to 2000 Pa, with no significant difference between the two stages (Figure 3B). When probing fresh pancreatic tissues, we assumed that we mostly encountered the extracellular matrix of the micro-dissected tissue. However, we probably also encountered surrounding pancreatic mesenchymal cells or fibroblasts, as suggested by the non-uniform distribution of elasticity in maps at both stages (Figure 3D). The images in Figure 3D are not clear. Authors are requested to provide clean and clear AFM images to explain the context appropriately.

 7.      The indentation experiments showed no noticeable difference in the biomechanical properties of the two selected stages. Henceforth, do the authors recommend ruling out this study in future research? If so, why?

 8.      Authors are requested to consult the following references. There are plenty of references on this subject area available. The claim “Lines 552-553: we provide the first-ever assessment of the viscoelasticity properties of the embryonic pancreas and its ECM, demonstrating a relative soft and low viscous matrix during development” can be justified appropriately.

References https://pubmed.ncbi.nlm.nih.gov/32984301/

https://www.sciencedirect.com/science/article/abs/pii/S1742706117307304

https://faseb.onlinelibrary.wiley.com/doi/abs/10.1096/fj.202200807R

 9.      The conclusions of the study need to be rephrased as it looks very weak.

10.  I recommend a minor revision of the manuscript.

Author Response

Our answers are in blue.
